# Neutrophil and Natural Killer Cell Interactions in Cancers: Dangerous Liaisons Instructing Immunosuppression and Angiogenesis

**DOI:** 10.3390/vaccines9121488

**Published:** 2021-12-16

**Authors:** Maria Teresa Palano, Matteo Gallazzi, Martina Cucchiara, Andrea De Lerma Barbaro, Daniela Gallo, Barbara Bassani, Antonino Bruno, Lorenzo Mortara

**Affiliations:** 1Laboratory of Innate Immunity, Unit of Molecular Pathology, Biochemistry, and Immunology, IRCCS MultiMedica, 20138 Milan, Italy; 2Laboratory of Immunology and General Pathology, Department of Biotechnology and Life Sciences, University of Insubria, 21100 Varese, Italy; matteo.gallazzi@uninsubria.it (M.G.); mcucchiara@studenti.uninsubria.it (M.C.); lorenzo.mortara@uninsubria.it (L.M.); 3Laboratory of Comparative Physiopathology, Department of Biotechnology and Life Sciences, University of Insubria, 20145 Varese, Italy; a.delermabarbaro@uninsubria.it; 4Endocrine Unit, Department of Medicine and Surgery, University of Insubria, ASST dei Sette Laghi, 21100 Varese, Italy; dgallo@hotmail.it; 5Molecular Immunology Unit, Department of Research, Fondazione IRCCS Istituto Nazionale dei Tumori, 20133 Milan, Italy; barbara.bassani@istitutotumori.mi.it

**Keywords:** neutrophils, natural killer cells, neutrophil-NK cell crosstalk, immunosuppression, tumor angiogenesis, tumor microenvironment

## Abstract

The tumor immune microenvironment (TIME) has largely been reported to cooperate on tumor onset and progression, as a consequence of the phenotype/functional plasticity and adaptation capabilities of tumor-infiltrating and tumor-associated immune cells. Immune cells within the tumor micro (tissue-local) and macro (peripheral blood) environment closely interact by cell-to-cell contact and/or via soluble factors, also generating a tumor-permissive soil. These dangerous liaisons have been investigated for pillars of tumor immunology, such as tumor associated macrophages and T cell subsets. Here, we reviewed and discussed the contribution of selected innate immunity effector cells, namely neutrophils and natural killer cells, as “soloists” or by their “dangerous liaisons”, in favoring tumor progression by dissecting the cellular and molecular mechanisms involved.

## 1. Introduction

### 1.1. Brief Overview on Neutrophils and Cancers 

Neutrophils are the most abundant circulating leukocytes, accounting for 50–70% of blood cells. During neutrophil maturation from common myeloid progenitors, the granulocyte-monocyte progenitor cells are the first to acquire the neutrophil lineage marker CD66b [1]. Then, during maturation, from promyelocytes to human mature neutrophils, the simultaneous acquisition of CD11b and CD16 can be observed [2,3] (Figure 1A). 

Neutrophils represent the first-line defense against infections, thus they are rapidly recruited from the bloodstream to the site of injury where they eliminate pathogens, particularly bacteria, by phagocytosis, degranulation, and release of extracellular traps (NETs) [4,5,6,7].

Although the functional diversity of neutrophils has been largely overlooked, compared with other myeloid cells, several studies have recently pointed out a high heterogeneity among them, also highlighting their plasticity [8]. It has been shown that even if neutrophil activation induces damage to the surrounding tissues, through the radical oxygen species (ROS) release, proteolytic enzymes, and antimicrobial proteins, they are also crucial for tissue regeneration, being able to phagocyte debris, to produce growth factors and pro-angiogenic proteins that promote re-vascularization, and to induce macrophage recruitment, which in turn supports and accelerates tissue repair [9]. These opposite functions, already described for other myeloid cells, such as macrophages, also suggests the existence of opposing polarization states (pro-tumoral or anti-tumoral) of neutrophils in cancer [10,11,12]. In line with this hypothesis, Zhu et al., identified seven distinct subsets of circulating neutrophils that differ in the markers’ expression, phagocytic abilities, ROS production, and most importantly, they correlate with melanoma stages, pointing out the clinical relevance of neutrophil heterogeneity in cancer patients [3]. Likewise, Ballesteros and colleagues identified different subsets of neutrophils across multiple tissues expressing peculiar receptors, displaying diverse transcriptional activity and chromatin features, characterized by distinct activities including vascular repair and hematopoietic homeostasis [13]. 

In cancer, mimicking the dichotomy between pro-inflammatory/M1-like and anti-inflammatory/tissue-repairing/TAM/M2-like macrophages, neutrophils have been conventionally termed N1 and N2 neutrophils, with the latter characterized by immunosuppressive, pro-metastatic, and pro-angiogenic activities [2,7,14]. 

Several studies support the pro-tumoral properties of N2 neutrophils, that predominantly express CC and CXC chemokines such as CCL2, 3, 4, 8, 12, and 17 and CXCL1, 2, 8, and 16, in addition to higher expression of vascular endothelial growth factor (VEGF), CXCR4 and matrix metalloproteinase 9 (MMP9) [14,15,16] (Figure 1B). Molecularly, one of the first mechanisms through which neutrophils promote tumor growth is represented by the release of ROS, which in turn induce DNA damage, which are essential for cancer initiation, cell proliferation, and increasing the mutational load [14,15,16]. 

Moreover, several studies [17] have shown that NETs produced by tumor-associated neutrophils (TANs) can degrade the extracellular matrix, promoting cancer cell extravasation and metastasis. Additionally, Albrengues et al., demonstrated the key role of extracellular traps in tumor relapse, showing that NET-associated proteases (i.e., elastase and MMP9 by laminin cleavage) promote the proliferation of dormant cancer cells [18].

Regarding the anti-tumor potential of N1 cells, it has been shown that “cytotoxic neutrophils” are characterized by TNFα^high^, CCL3^high^, ICAM1^high^, and arginase^low^ expression [19,20] (Figure 1C) and can mediate anti-tumor functions through the production of TNF-related apoptosis-inducing ligand (TRAIL), which has been shown to induce apoptosis in Jurkat leukemic cells [21]. Moreover, Blaisdell and colleagues demonstrated that neutrophils promote tumor cell detachment from the basement membrane, inhibiting the initial phases of carcinogenesis [22]. Accordingly, Mahiddine et al., showed that, although an hyperoxic microenvironment limits neutrophil recruitment within tumors, infiltrated neutrophils displayed a high anti-tumor potential, due to the increased release of MMP9 and ROS, which limit tumor cell proliferation and induce apoptosis [23]. Although neutrophils can exert direct anti-tumor or pro-tumoral functions, the regulatory role of these cells in orchestrating other cells of the tumor microenvironment (TME) of both myeloid and lymphoid origin has clearly been stated [24]

### 1.2. Brief Overview on NK Cells and Cancers 

Natural killer (NK) cells are large granular lymphocytes from innate immunity, participating in the early recognition and eliminations of virus-infected and malignant-transformed cells. Currently, NK cells are classified as innate lymphoid cells (ILCs) and originate from common innate lymphoid progenitors in the bone marrow. Subsequently, they migrate to different lymphoid tissues or non-lymphoid tissues, where they are “educated”, acquiring phenotype and functions typical of tissue residency. The NK inhibitory receptor repertoire is adapted to the MHC class I molecules borne by the host, assuring NK cell tolerance against self-cells. On the other hand, NK cells are simultaneously stimulated by activating receptors that trigger NK cell responses. In the presence of healthy cells, activating signals are low, thus the binding of inhibitory receptors to MHC class I molecules is sufficient to induce NK cell tolerance. In contrast, when NK cells recognize altered cells (i.e., tumor cells) that lack or downregulate MHC I expression, the activating signals overcome the inhibitory ones, leading to the killing of altered cells [25]. 

Based on their ability to express the CD56 (neural cell adhesion molecule-NCAM) and CD16 (Fc receptors FcγRIII, involved in the antibody-dependent cellular cytotoxicity (ADCC) process) surface antigens, two major NK cell subsets have been characterized. CD56^dim^CD16^+^ (more mature phenotype) NK cells account for 90–95% of circulating NKs and exert their killing function by releasing perforin, granzymes, and mediating ADCC (Figure 1C). CD56^bright^CD16^−^ NKs (considered less mature NKs) account for 5–10% of circulating NKs and act both as distant killers and regulatory cells, by producing TNFα and IFNγ (Figure 1D), which can exert direct anti-tumor activity, or can activate other effector cells of both innate and adaptive (T cell mediated) immunity [26].

A peculiar immunosuppressive and pro-angiogenic NK cell subset has been characterized within the developing decidua, termed decidual NK cells (dNKs), which are identified as CD56^superbright^ CD16^−^ cells and support the formation of the spiral arteries, by producing VEGF, placental growth factor (PlGF), and CXCL8 (IL8) (Figure 1D). dNK cells represent a clear scenario of NK cell plasticity and phenotypic/functional adaptation, demonstrating how cells generated as “killers” can be turned into “builders”, as a consequence of a specific organism demand.

Even if little NK-cell infiltration has been reported in solid tumors, NK cell infiltration has been observed in several types of cancers, including melanoma [27], gastro-intestinal stromal tumors (GIST) [28], colorectal [29], renal [30], lung [31], and breast cancers [32]. 

Several studies demonstrated a relevant role for NK cells in cancer progression and patient outcome [33,34]. Despite initial studies showing that cancer patients displaying high numbers of tumor infiltrating NK cells are characterized by a better prognosis and reduced metastasis [35,36], recent studies have shown that tumor incidence in NK cell-deficient patients was not higher than in healthy subjects, suggesting a compensatory mechanism due to the activation of other leukocytes [37,38]. Moreover, NK cells infiltrating solid cancers have been reported to be predominately CD56^bright^ with impaired cytotoxic functions [39,40,41]. Several mechanisms within the TME can alter NK-cell functions including: (i) decreased expression of the natural cytotoxicity receptors (NCRs) NKp30, NKp44 and NKp46, together with reduced levels of the NKG2D activation marker and impaired degranulation capabilities; (ii) increase of cell exhaustion markers, such as NKG2A, PD-1, TIGIT, and CD96 molecules; and (iii) NK cell inhibition by both extracellular stimuli such as hypoxia, transforming growth factor β (TGFβ) production, prostaglandin-E2 secretion, and immunosuppressive cells such as MDSCs and Treg cells [42,43,44]. Interestingly, pro-inflammatory, pro-angiogenic, and decidual-like NK cells have been characterized in the peripheral blood and tissue of patients with NSCLC, CRC, PCa and in the pleural effusions of metastatic cancer patients, defined by a CD56^bright^CD9^+^CD49a^+^ phenotype [41,45,46,47]. Additionally, these dNK-like cells produce VEGF, PlGF, CXCL8 [41,46], angiogenin [46], and MMPs [46], which can support inflammatory angiogenesis ex vivo [41,45,46,47] and polarize macrophages toward M2-like/TAMs [45]. Mechanistically, TGFβ [45] and the STAT3/STAT5 [46] axis have been described as major drivers of the NK cell angiogenic-switch in solid cancer patients.

In line, Gotthardt et al., highlighted the relationship between NK cells and angiogenesis, demonstrating that Stat5∆/∆Ncr1-iCreTg-Vav-Bcl2 mice failed to control tumor growth of RMA-S lymphoma cells and a v-abl transformed tumor, clarifying the significant tumor-promoting function of STAT5-deficient NK [48]. Furthermore, Guan et al., showed in renal cancer patients that NK cells are switched toward a pro-angiogenic/dNK phenotype, characterized by impaired cytotoxicity, along with increased ability to produce VEGF and an enrichment in hypoxia-related genes [49]. Therefore, as for neutrophils, NK cells are high plastic cells able to adapt their phenotype and functions in response to different stimuli within TME. Interestingly, it has been shown in both humans and murine models, that mature neutrophils are required for proper NK cell development [50]. In a mouse model (Genista) characterized by neutrophil deficiency, NK cells displayed impaired maturation, function, and homeostasis [50].

This evidence suggests that bi-directional interplay between NK cells and neutrophils act as a relevant hub in shaping the TME and the possible response to specific (immune)therapies, finally placing this dangerous liaison as a new potential immune-related hallmark of cancer.

## 2. Neutrophil–NK Cell Interactions: A Dangerous Liaison in the Immunosuppressive TME 

Several studies have demonstrated that TANs can directly or indirectly (via crosstalk with other immune cells) contribute to the generation of an immunosuppressive TME. Studies of neutrophil-induced immunosuppression have mostly focused on their ability to inhibit T cell functions [24,51]. Here, we focused on neutrophil-NK cell interactions as a critical step in the immunosuppressive TME.

In a colorectal cancer murine model, generated by CT-26 cells intramuscularly injected into the flanks of BALB/c mice, neutrophils have been shown to suppress the NK cell infiltration, by downregulating CCR1 and to impair anti-tumor capabilities (Figure 2A) by cell-to-cell interactions, through the PD-L1/PD-1 axis [52] (Figure 2B). 

Neutrophils isolated from the CT-26 tumor-bearing mice, when co-cultured both with naïve and tumor-bearing NK cells, displayed a decreased production of IFNγ; treatment with PD-L1 neutralizing antibody was effective in limiting tumor-bearing neutrophil inhibitory effects on tumor-bearing NK cells, but did not exert the same inhibitory effect on naïve NK cells [52]. Additionally, administration of PD-1 neutralizing antibodies was able to antagonize the inhibitory effect of neutrophils on NK cells [52].

The NKp46 NCR is considered a key molecule for NK-related killing capacity; Valayer et al., showed that NKp46 decreasing on NK cells is related to activated neutrophil-derived serine proteases and in particular, cathepsin G (CG) is responsible for NKp46 extracellular cleavage [53] (Figure 2Ci), thus causing defective activation of NK cells in in vitro experiments using human-derived cells. The use of a specific CG inhibitor, α1ACT, abrogated the capability of neutrophil-derived conditioned media to decrease NKp46 on the NK cell surface [53]. 

Neutrophils can activate or suppress the NK cells cytotoxic functions and therefore can indirectly interfere with the angiogenic process. Romero et al., demonstrated that neutrophil-derived ROS can modulate NKp46 expression on NK targeted cells [54] (Figure 2Cii). Indeed, neutrophil-derived ROS downmodulated NKp46 in CD56^dim^ NK cells [54] (Figure 2Cii), while the opposite effect was exerted in CD56^bright^ NK cells, in which NKp46 was increased, probably due to high anti-oxidative intrinsic capability [54] (Figure 2Cii), and this modulation is revered by catalase [54]. Thus, neutrophil-derived ROS, by NKp46 modulation of NK cells, can enhance or reduce their cytotoxic effect against endothelial cells (ECs), as reported by Dondero and colleagues, that showed that NKp46 is involved in ECs killing in multiple myeloma (MM). Similar to ROS and CG, elastase and lactoferrin act on NK cells, increasing their cytotoxicity [55] (Figure 2Ciii). Arginase I (ARG1), released by TANs, also participate in supporting NK cell pro-angiogenic features, suppressing NK cell capability to produce anti-tumor factors such as IFNγ [56] (Figure 2Di).

IL12 is also produced by neutrophils and is crucial for optimal IFNγ and perforin production by both murine and human NK cells [57]. IL12 signaling, through STAT4 activation, induces IFNγ production in NK cells, as confirmed in a mouse model lacking Stat4, in which NK cells display lower IFNγ production and thus decreased cytolytic function [58] (Figure 2Dii). The same mechanism occurs in humans, as shown by using the human NKL cell line in which STAT4 activation by IL12 is directly related to perforin expression in in vitro experiments [59]. Moreover, IFNγ can exert its effect on TANs. As observed in C57BL/6 mice implanted with the MCA205 murine fibrosarcoma cell line, NK cell depletion and IFNγ deficiency allowed for an increased tumor growth compared to the control mice [60]. This effect is mediated by TANs, thus the absence of NK cells increased the pro-angiogenic features of TANs [60] (Figure 2E).

Neutrophil and NK cell interactions have been reported to be crucial in supporting the metastatic process. Spiegel et al., using a murine model (BALB/c mice) injected with 4T1 mammary carcinoma cells, showed that neutrophils can suppress intraluminal NK-mediated tumor cell elimination and enhance extravasation of disseminated carcinoma cells [61].

Using E0771 tumor bearing mice, Li et al., reported that neutrophils either promote or inhibit metastasis, depending on the presence of NK cells [62]. They showed that following granulocyte colony-stimulating factor (G-CSF)-induced neutrophil expansion, both immunocompetent and NOD-*scid* mice were more prone to develop lung metastases, whereas NOD-*scid IL2rγ^null^* mice, which also lack NK cells, are characterized by a reduced number of metastases [62]. Conversely, by producing ROS, neutrophils can either suppress the tumoricidal activity of NK cells or mediate cancer cell killing [62].

## 3. Neutrophil–NK Cell Interactions: Contribution to Tumor Angiogenesis

Neutrophils represent one of the first infiltrating cell type within TME and can shape TME promoting tumor growth and metastasis formation by angiogenesis stimulation [8,63]. In a skin model of wound healing in CD18-deficient mice that lack neutrophils, it has been shown that neovascularization is compromised compared to wild-type animals [64], suggesting that neutrophil infiltration ameliorates/improves angiogenesis.

Angiogenesis stimulation can be mainly mediated by the release of soluble factors including the main pro-angiogenic master regulator VEGF [65,66]. Vessel-associated neutrophils are a key source of VEGF upon stimulation with CXCL1 [67] and G-CSF [68] are one of the main producers of MMP9, which contributes to angiogenic switch in Rip-tag pancreatic cancer in the in vivo mouse model, where neutrophil depletion at the early stage inhibits angiogenesis progression [69]. In addition, through gene expression analysis, Schruefer and colleagues identified novel pro-angiogenic factors produced by human neutrophils such as ephrin A2 and B2, thrombospondin, TGFβ receptor 2 and 3 (TGFβR2 and TGFβR3), tissue inhibitor of metalloproteinase 2, and restin [64]. Conversely, neutrophils can also produce anti-angiogenic [70] factors as they can release enzymatic activities that in vitro generate active angiostatin fragments that in turn inhibit basic fibroblast growth factor (bFGF/FGF2) and VEGF-induced EC proliferation [70], thus negatively interfering with the angiogenic process. On ECs, VEGF can increase ICAM-1 and VCAM-1 expression, which in turn, can respectively mediate neutrophil and NK cell recruitment by interaction with CD18 expressed by both neutrophils and NK and CD49d on the NK cell surface [66,71,72].

NK cells can play a dual role related to angiogenesis by acting as the inhibitor and promoter of this process. In response to IL12 [71], cytotoxic NK cells start producing IFNγ [72], which exerts an anti-angiogenic effect through IFNγ inducible protein-10 (IP-10) accumulation [73] (Figure 3A). 

Indeed, IL12 increased the production of cytolytic mediators as granzyme and perforin by NK cells and blocked the pro-angiogenic effect of FGF2 by IP-10 (CXCL10) (Figure 3A), causing tumor necrosis and vascular damage in experimental Burkitt lymphomas. Interestingly, IP-10/CXCL10, which sustains the NK cells’ anti-angiogenic activity, can also be produced by neutrophils upon IFNα stimulation [74], thus affecting both NK and neutrophil behavior.

The anti-angiogenic loop that involves IP-10 can be driven by both NK cells and neutrophils that together can contribute to angiogenesis inhibition. The anti-angiogenic effect of IFNγ can also be boosted by other neutrophil-derived cytokines as IL12, IL15, and IL18 [75,76] (Figure 3A) that, due to their pro-inflammatory function, activates NK cells and increases NK-derived IFNγ, thus sustaining angiogenesis inhibition [75,76] as also shown by an in vitro coculture experiment in which neutrophils enhanced NK production of IFNγ [77].

CXCL8 can work as a bridge-molecule between NK cells and neutrophils. CXCL8 can target neutrophils, increasing their recruitment, which in turn, sustains angiogenesis together with tumor progression in Ras oncogene driven tumors [78]. In addition, in vitro coculture of NK cells and neutrophils increased neutrophil CXCL8 production [77], corroborating the crosstalk between NK and neutrophils.

Activated NK cells increased CD11b expression on neutrophils and promoted CD62L (L-selectin) shedding through IFNγ and granulocyte-macrophage colony stimulating factor (GM-CSF) release [79] (Figure 3B). In a transplantable tumor mouse model obtained with subcutaneous injection of melanoma and fibrosarcoma cells, Jablonska and colleagues have shown that CD11b^+^ neutrophils are responsible for angiogenesis stimulation since expressed high levels of genes encoding for VEGF and MMP9, together with transcription factor c-myc and STAT3, which are positive regulators of both VEGF and MMP9 [79] (Figure 3B). In addition to CD11b, NK modulated neutrophil expression of CD62L and CD64. In detail, NK-derived IFNγ and GM-CSF promoted CD64 expression [80] (Figure 3B). 

In a work published by Romano et al., CD64 expression on neutrophils correlated with immune-suppression and tumor progression [81]. Studying MM patients compared with healthy donors and MGUS (monoclonal gammopathy of undetermined significance, early stage of myeloma) subjects, Romano and colleagues showed that CD64 expression on neutrophils increased from healthy to MGUS and to MM patients together with p-STAT3 [81]. The concomitant increase in CD64 expression on neutrophils during myeloma progression could be linked with angiogenesis promotion, considering that MM-neutrophils displayed a N2-like phenotype with pro-angiogenic features. In addition, CD64-expressing neutrophils showed increased ARG1 expression together with p-STAT3 during myeloma progression. As mentioned before, ARG1 [56] could play a positive role for angiogenesis and, in addition, STAT3 could increase pro-angiogenic VEGF and MMP9 production [79] (Figure 3B), therefore CD64-expressing neutrophils can contribute to angiogenic switch in MM.

In contrast, in a sarcoma transplantable model, the effect of NK-derived IFNγ regulated neutrophil function, preventing/impeding their pro-angiogenic activity [82]. Ogura and colleagues showed that IFNγ and NK cells negatively controlled neutrophil-derived VEGF-A [60,83] (Figure 3C). Indeed, the supernatant from IFNγ-KO mice switched on VEGF-A mRNA expression in neutrophils [60,83]. This modulation involved TME as VEGF-A protein expression is enhanced in NK-depleted mice. Finally, by the Matrigel plug angiogenesis assay with the MCA205 murine fibrosarcoma cell line, Ogura further confirmed the link between NK and neutrophils in angiogenesis control since vascularization is enhanced in the absence of NK cells (obtained with anti-asialo GM1 antibody) and this process is neutrophil-dependent since in neutrophil depleted mice (using anti-Ly6G antibody), angiogenesis is reduced as in the control [60,83] mice [60].

## 4. Conclusions

Immune cell plasticity can be envisaged as a relevant host-dependent hallmark of cancers, characterized by tumor-infiltrating and tumor-associated immune cell capability to acquire tumor-supporting phenotypes and functions. In this scenario, it is now clear that not only the immune cell phenotype/functional switch (pro-tumor/pro-angiogenic/pro-metastatic) but also the dangerous liaisons occurring between different immune cells in the tumor micro- and macro-environments, represent crucial events impacting cancer progression and the success of the therapeutic regimens. Based on these concepts, immunotherapy has emerged as the new next-generation approach in cancer therapy, with different success in some cancer types (melanoma, lung cancer) and with minor or null success in other cancer types. The still persistent window of “non-successful” immunotherapy further opens the need to deepen the investigation into the immune cell–TME interactions in cancer. This is a crucial point in perspective/future immunotherapeutic approaches, where modifying/re-educating the immune system should take into consideration not only the action on altered immune cells such as soloist elements, but also on the immune cells/TME interactions.

## Figures and Tables

**Figure 1 vaccines-09-01488-f001:**
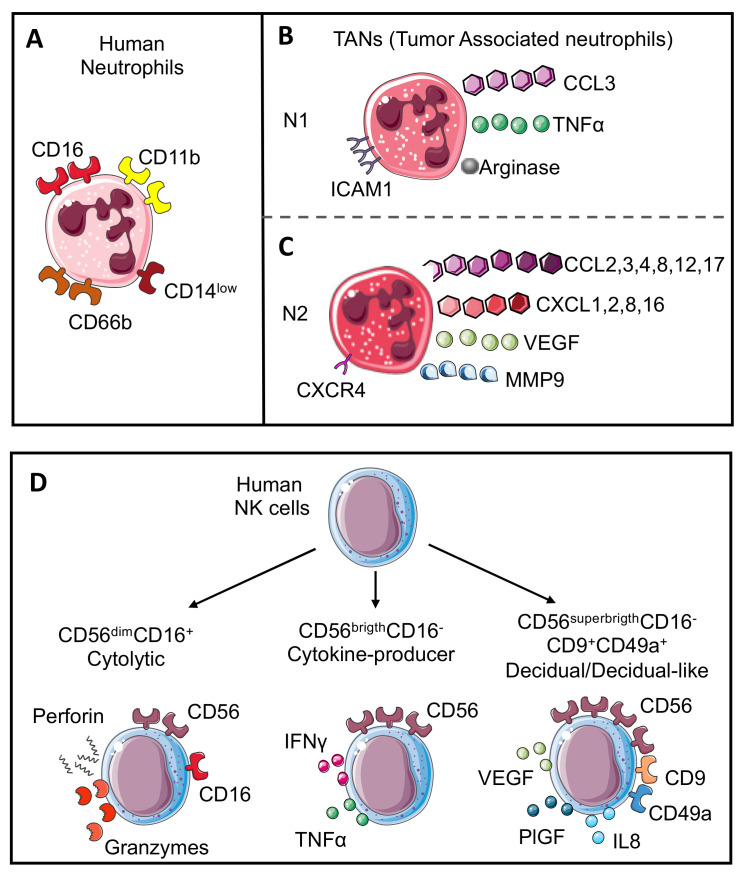
**Neutrophil and NK cell subsets**. Human neutrophils (**A**) are characterized by CD11b, CD66b, CD14, and CD16 expression. Mimicking the M1 or M2-like macrophage polarization, neutrophils in the tumor microenvironment are also characterized by the capability to acquire two opposite phenotypes. N1 neutrophils (**B**) show anti-tumoral activities and are characterized by the expression of chemokine (C-C motif) ligand 3 (CCL3), tumor necrosis factor α (TNFα), arginase, and intercellular adhesion molecule 1 (ICAM1); in contrast, N2 neutrophils (**C**) work as tumor-promoters and express CC and CXC chemokines (CCL2, 3, 4, 8, 12 and 17 and CXCL1, 2, 8, and 16), vascular endothelial growth factor (VEGF), matrix metalloprotease 9 (MMP9), and CXCR4 receptor. Human NK cell subsets (**D**) are characterized by CD56 expression. Cytolytic NK cells express CD56 and CD16 (CD56^dim^CD16^+^) and can release perforin and granzyme; cytokine producer NK cells lose CD16 expression and increase CD56 expression (CD56^brigth^CD16^−)^ with the production of cytokines including TNFα and interferon γ (IFNγ); the last subset is named decidual cells (dNKs) that displayed higher expression of CD56 with CD9 and CD49a markers (CD56^superbrigth^CD16^−^CD9^+^CD49a^+^) and support angiogenesis through the release of VEGF, placental growth factor (PlGF), and interleukin 6 (IL6). A similar subset named decidual-like NK cells is also found in the tumor microenvironment (TME).

**Figure 2 vaccines-09-01488-f002:**
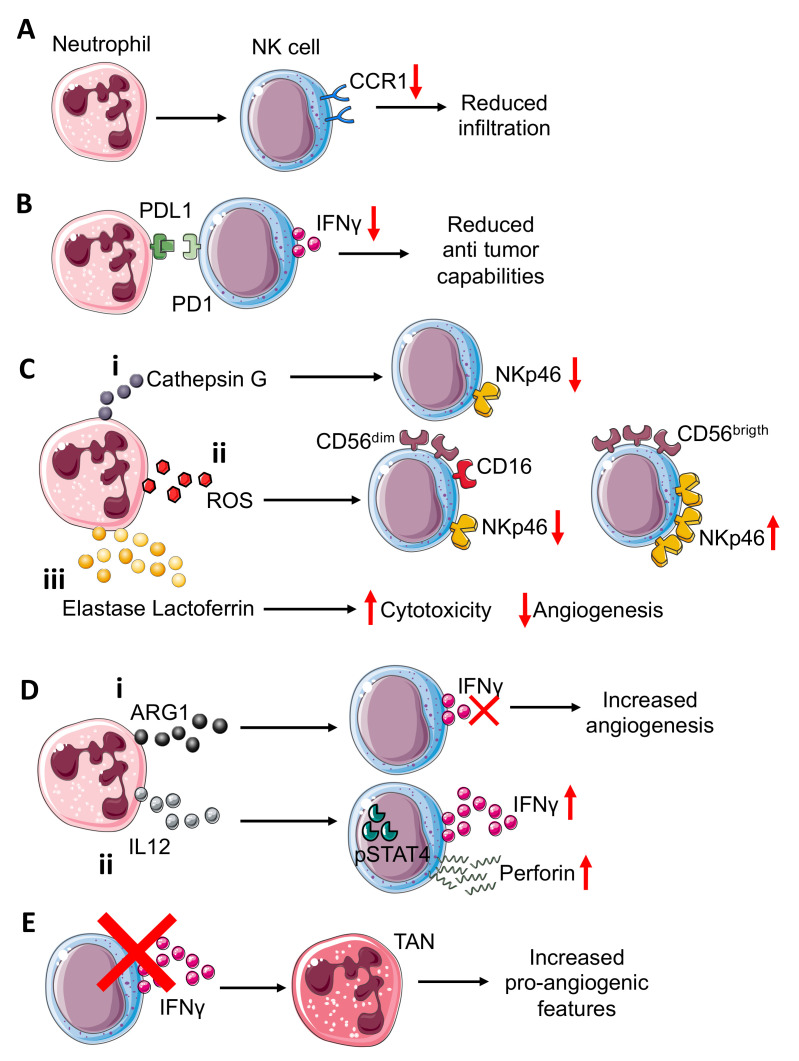
**Neutrophil–NK cell crosstalk in tumor microenvironment**. Neutrophils can (**A**) induce reduction in CCR1 expression on NK cells, impairing the NK cells’ infiltration capability. Interference with PDL1-PD1 (**B**) interactions in the TME, resulting in reduced NK cell capability to release IFNγ. Neutrophils can also modulate the expression of activating receptor NKp46 on NK cells through (**C**) the release of different molecules that include: (i) neutrophil-derived Cathepsin G (CG) reduces NKp46 on NK cells; similar; (ii) reactive oxygen species (ROS) can downmodulate NKp46 on cytotoxic CD56^dim^CD16^+^NK cells while they can upregulate this receptor on cytokine-producer CD56^brigth^CD16^−^ NK cells; (iii) elastase and lactoferrin release exert a wide effect increasing cytotoxicity and reducing angiogenesis. NK-derived IFNγ, a key mediator in TME, can be inversely modulated by ARG1 and IL12 from neutrophils (**D**). (i) Indeed, neutrophil-derived ARG1 can abrogate IFNγ released from NK, improving NK pro-angiogenic features; (ii) while neutrophil-derived IL12 through STAT4 activation increases IFNγ and perforin production by NK cells. NK cell–neutrophil crosstalk (**E**) can be modulated by NK-derived IFNγ, which acts by decreasing pro-angiogenic features of tumor associated neutrophils (TANs). Indeed, NK cells or IFNγ depletion increase TANs’ pro-angiogenic features.

**Figure 3 vaccines-09-01488-f003:**
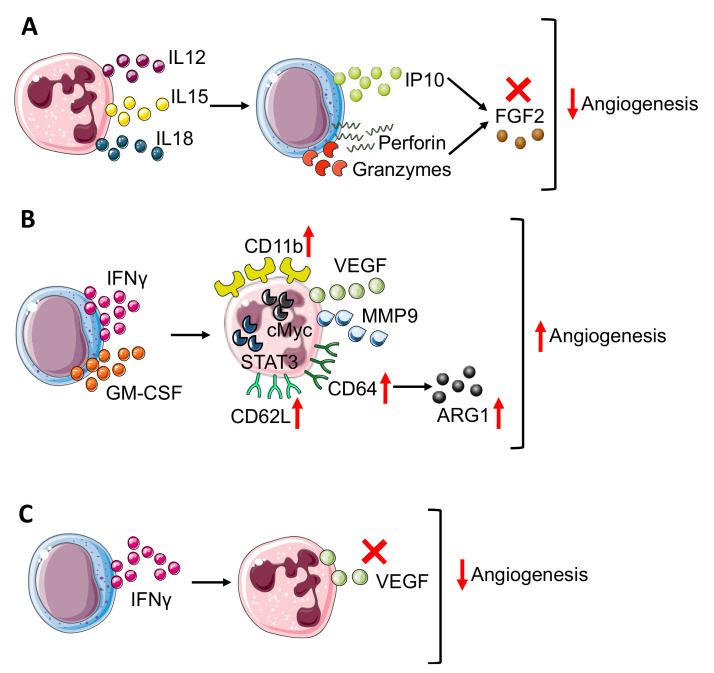
**Neutrophil–NK cell crosstalk in angiogenesis modulation**. The bidirectional crosstalk between neutrophils and NK cells can negatively modulate angiogenesis thus (**A**) neutrophil derived IL12, 15, and 18 stimulate IP10, perforin, and granzyme production in NK cells that block FGF2 effect. In TME, NK cells can improve angiogenesis (**B**) through the production of IFNγ and granulocyte-macrophages colony-stimulating factor (GM-CSF), which results in neutrophil expression of CD11b, CD62L, and CD64 surface antigens and release or VEGF and MMP9, in a STAT3 and c-Myc dependent manner. Moreover, in neutrophils, CD64 expression is linked to increasing production of ARG1. On the other hand, (**C**) NK-derived IFNγ prevents VEGF release from neutrophils, reducing angiogenesis stimulation. Finally, panels B and C show how IFNγ, produced by NK cells, act as a double edge sword by exerting both pro-angiogenic and anti-angiogenic activities.

## Data Availability

Not applicable.

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
