# Peer review of "Neutrophil and Natural Killer Cell Interactions in Cancers: Dangerous Liaisons Instructing Immunosuppression and Angiogenesis"

_vaccines, 2021, doi:10.3390/vaccines9121488_

Round 1

Reviewer 1 Report

The study aims to describe the role of the immune microenvironment and its influence in tumor progression. Neutrophils and Natural Killer cells and their role in tumor progression are described. It is therefore necessary to re-educate the host's immune system. The authors have already published other works on this important topic useful for new anti-cancer approaches. The review is well written but there are some things that should, in my opinion, be changed to make reading more fluid.

1) Introduction: The general part on neutrophils and NK cells is too long.

The authors should emphasize the importance of the interactions between neutrophils and NK cells because these are the most important chapters. In my opinion these points are essential for the study.

2) Figures: the authors should replace figure 1 (specially A and B ) with one or two less confusing figures.

-the A in figure 2 is missing

-replace "bare" with "are" in the legend of Figure 1

Author Response

The study aims to describe the role of the immune microenvironment and its influence in tumor progression. Neutrophils and Natural Killer cells and their role in tumor progression are described. It is therefore necessary to re-educate the host's immune system. The authors have already published other works on this important topic useful for new anti-cancer approaches. The review is well written but there are some things that should, in my opinion, be changed to make reading more fluid.

We thank the reviewer for these positive comments.

1) Introduction: The general part on neutrophils and NK cells is too long.The authors should emphasize the importance of the interactions between neutrophils and NK cells because these are the most important chapters. In my opinion these points are essential for the study.

We agree with the reviewer suggestions  and revised the test accordingly.

2) Figures: the authors should replace figure 1 (specially A and B ) with one or two less confusing figures.

-the A in figure 2 is missing

-replace "bare" with "are" in the legend of Figure 1

We revised figure 1 to make it more easy to follow within the organization/logic of the specific figures. Typos have been checked and revised.

Reviewer 2 Report

This is a review article focused on the role of neutrophils and natural killer (NK) cells as well as of their interactions in tumor biology. In the first part of the paper authors present the general characteristics of neutrophils and NK cells and their role in tumor microenvironment. In the second part of the paper authors focus on the interaction between both cell types in the context of immunosuppression and angiogenesis.

The topic is of interest and the manuscript is well-written. A lot of data from original studies are presented. The text is supported by comprehensive informative figures. References are correctly formatted and up to date. I have no critical comments.

Author Response

We thank the reviewer for the posive comments.

Reviewer 3 Report

This is an interesting review aiming to reveal the interaction of NK cell and neutrophiles in tumorigenesis. The authors provided a line of evidence to describe the controvery relationship of these cells on the tumor development. Overal, this review is useful for readers to understand the progress of the field. However, the manuscript should be revised before its publication.

  1. “it can be observed the 40 concomitant acquisition of CD11b and CD16”is confusing.
  2. “Human neutrophils (A)bare characterized by the surface 43 antigen espression of CD11b, CD66b, CD14, CD16” is confusing.
  3. At line 268, “Similarly to ROS and CG” should be “Similar to ROS and CG”.
  4. At line 323, “Interesting” should be “interestingly”.
  5. More literatures should be cited. For instance, “Dual roles of neutrophils in metastatic colonization are governed by the host NK cell status. Nat Commun, 2020”; “Neutrophils Suppress Intraluminal NK Cell-Mediated Tumor Cell Clearance and Enhance Extravasation of Disseminated Carcinoma Cells. Cancer Discov. 2016."

Author Response

We thank the reviewer for the suggestions  received.

Point 1-4: we revised the text according to the reviewer's suggestions.

Point 5: we really appreciate the suggestions by the reviewer and we revised accordingly, also citing the relevant papers indicated.

Round 2

Reviewer 1 Report

I can accept the paper in this form.

Reviewer 3 Report

The authors revised the artical well and I agree its publication.